# HPV and Lung Cancer: A Systematic Review

**DOI:** 10.3390/cancers16193325

**Published:** 2024-09-28

**Authors:** Telma Sequeira, Rui Pinto, Carlos Cardoso, Catarina Almeida, Rita Aragão, Teresa Almodovar, Manuel Bicho, Maria Clara Bicho, Cristina Bárbara

**Affiliations:** 1Serviço de Pneumologia, Instituto Português de Oncologia (IPO), Rua Lima Basto, 1099-023 Lisboa, Portugal; katarynalmeida@hotmail.com (C.A.); aragaorita@hotmail.com (R.A.); m.teresaasa@gmail.com (T.A.); 2Laboratório Associado TERRA, Instituto de Saúde Ambiental (ISAMB), Faculdade de Medicina, Universidade de Lisboa, Av. Professor Egas Moniz, 1649-028 Lisboa, Portugal; mcbicho@medicina.ulisboa.pt (M.C.B.); cristina.barbara@ulssm.min-saude.pt (C.B.); 3Joaquim Chaves Saúde, Rua Aníbal Bettencourt, n° 3, Edifício CORE, 2790-225 Oeiras, Portugal; rui.pinto@jcs.pt (R.P.); carlos.cardoso@jcs.pt (C.C.); 4Instituto de Investigação Científica Bento da Rocha Cabral, Calçada Bento da Rocha Cabral 14, 1250-012 Lisboa, Portugal; manuelbicho@medicina.ulisboa.pt; 5Unidade Local de Saúde de Santa Maria, Av. Professor Egas Moniz, 1649-028 Lisboa, Portugal

**Keywords:** HPV, lung cancer, adenocarcinoma, squamous cell carcinoma

## Abstract

**Simple Summary:**

Lung cancer remains a significant global health challenge with high incidence and mortality rates worldwide. In recent years, there has been a growing recognition of the role of Human Papillomavirus (HPV) in lung cancer development. This systematic review aims to explore the diagnostic criteria, epidemiology, etiology, and prognosis of HPV infection in lung cancer. A total of 97 studies encompassing 9098 patients worldwide, revealing varied HPV infection rates in lung cancer, ranging from 0% to 69%, were analyzed. While HPV16/18 was predominant in some regions, its association with lung cancer remained inconclusive due to conflicting findings. Some studies suggested a limited role of HPV in lung carcinogenesis, particularly in non-smokers. Despite inconclusive evidence, intriguing associations between HPV and lung adenocarcinoma and squamous cell carcinoma have emerged. Further research with standardized methodologies and larger cohorts is needed for clarity.

**Abstract:**

This systematic review aims to explore the diagnostic criteria, epidemiology, etiology, and prognosis of Human Papillomavirus (HPV) infection in lung cancer. This PRISMA-guided review searched the PubMed^®^ and Embase^TM^ databases for “lung cancer AND HPV” on 10 June 2023, filtering human subject papers. A total of 97 studies encompassing 9098 patients worldwide, revealing varied HPV infection rates in lung cancer, ranging from 0% to 69%, were analyzed. While HPV16/18 was predominant in some regions, its association with lung cancer remained inconclusive due to conflicting findings. Studies from Asia reported lower HPV infection rates compared to Western populations. Some studies suggested a limited role of HPV in lung carcinogenesis, particularly in non-smokers. However, intriguing associations were noted, including HPV’s potential role in lung adenocarcinoma and squamous cell carcinoma. Discrepancies in HPV detection methods and sample sources highlight the need for further research with standardized methodologies to elucidate HPV’s role in lung carcinogenesis and its clinical implications. Overall, this systematic review offers insights into HPV’s role in lung cancer epidemiology and clinical characteristics. Despite inconclusive evidence, intriguing associations between HPV and lung adenocarcinoma and squamous cell carcinoma have emerged. Further research with standardized methodologies and larger cohorts is needed for clarity.

## 1. Introduction

Lung cancer is the leading cause of cancer-related mortality worldwide, with an estimated 2 million diagnoses and 1.8 million deaths [1]. In particular, the incidence of lung cancer in non-smokers, predominantly women, has been increasing in recent years [2]. Several risk factors may contribute to lung cancer development in non-smokers, including secondhand smoke, indoor air pollution, occupational exposure, and genetic susceptibility. Of note, adenocarcinoma represents the most common histology in this population [2].

Human Papillomavirus (HPV) is a double-stranded DNA virus and is a representative virus known to be an etiologic agent in both benign and malignant tumors [3]. HPV infections cause up to 4.5% of all new cancers worldwide and represent 29.5% of all infection-related cancers [4]. Syrjanen was the first to report a case with condylomatous changes in neoplastic bronchial epithelium [5]. Additionally, HPV-6 and HPV-11 are involved in the formation of respiratory papillomas, with occasional malignant transformation of infected cells [6]. However, its role in lung cancer is still under investigation, originally supported by the fact that lung cancers had morphological similarities with anogenital cancers caused by HPV [7]. The molecular mechanisms of HPV-induced carcinogenesis are related to its major oncogenes, E6 and E7, which inactivate p53 and Rb, respectively. The HPV-E7 protein binds to and promotes the degradation and functional inactivation of Rb, leading to the release of E2F, which encodes for a family of transcriptional factors that regulates the cell cycle and induces G1/S cell cycle transition [8]. Negative feedback leads to P16 protein overexpression [9].

A 2009 meta-analysis [10] and a systematic review [11] evaluated the accumulated evidence and independently concluded that HPV may be a risk factor for some histologies of lung cancer. A 2015 meta-analysis reported an association between HPV infection and lung cancer [12], but others have raised concerns regarding the limited number of included studies and possible confounding [13]. In 2017, a meta-analysis suggested that HPV infection is a prognostic marker in lung adenocarcinoma. Nevertheless, to further elucidate the epidemiology and pathogenesis of HPV infections, future larger prospective studies are encouraged [14]. In 2021, the most recently published systematic review and meta-analysis about HPV and lung cancer showed that the HPV prevalence in lung cancer varies based on geographic localization, and HPV-16 and HPV-18 were the most prevalent high-risk genotypes identified. The authors emphasize that their review provides convincing evidence that HPV infection increases the risk of developing cancer [15]. However, manuscripts noted substantial heterogeneity in the reported data and argued that further studies were needed [15]. Of note, in the 2021 meta-analysis, only HPV PCR results from fresh-frozen and paraffin-embedded tissue were included, while in the present review, all HPV detection methods were considered. Furthermore, while the 2021 meta-analysis included a total of 78 selected publications, herein, a total of 97 studies were considered after the application of selection criteria. Thus, the main goal of this systematic review of the literature is to summarize the existing knowledge about the role of HPV infection in lung cancer development. This study is divided into three main topics: diagnostic criteria, epidemiology and etiology, and prognosis. This systematic review aims to detail the various diagnostic criteria used for identifying HPV infection and diagnosing lung cancer in the studies reviewed. It includes the methods used for HPV detection and the criteria for diagnosing lung cancer. It describes the epidemiological aspects related to HPV infection and lung cancer, including the prevalence of HPV in patients with lung cancer across different populations, regions, and lung cancer subtypes. It investigates the potential role of HPV as an etiological factor in the development of lung cancer, summarizes the existing evidence on the association between HPV infection and the risk of developing lung cancer, explores whether HPV infection acts independently or in conjunction with other risk factors in lung cancer development, analyses any variations in prevalence and trends over time, and assesses the prognostic implications of HPV infection in patients with lung cancer. It also reviews studies that evaluate whether HPV presence correlates with specific clinical outcomes, such as survival rates, tumor characteristics, responses to treatment, or disease progression.

## 2. Methods

This review adheres to the guidelines set by the Preferred Reporting Items for Systematic reviews and Meta-Analysis (PRISMA) [16]. This study was not registered in PROSPERO. The PubMed^®^ and Embase^TM^ databases were searched on 10 June 2023 using the strings “lung cancer AND HPV”. The search was restricted to papers involving human subjects. SLR and meta-analysis will only be used for reference cross-checking. Results from these searches were combined, and duplicate entries were removed. Two authors independently reviewed the abstracts for all publications identified, and in the case of discrepancies, a third author made the final determination regarding the publication for this review.

Inclusion criteria: All articles that included an adult population with lung cancer and HPV detection were included. Exclusion criteria: Studies that were not published in the English language and that did not assess the presence and/or a direct relationship between HPV and lung cancer were excluded. No date restrictions were applied. The data from each eligible study was extracted using a standardized data extraction form. This form includes key study characteristics, such as author name, publication year, sample size, geographic location, sociodemographic characteristics, histology type, smoking status, type of sample for HPV detection, HPV prevalence in lung cancer cases, and type of HPV. The potential bias of the studies was analyzed based on study design, population and sample, outcomes, data collection and analysis, and quality of the report.

## 3. Results

### 3.1. Evidence Search

A comprehensive flow chart with the results of the literature search is illustrated in Figure 1.

Initially, 440 records were exported from the PubMed and Web of Sciences databases. Following the elimination of duplicates and records lacking available abstracts, a total of 386 publications were retained. Of these, 289 were deemed ineligible for inclusion based on the application of the selection criteria. Detailed reasons for full-text exclusions and the article selection process are represented in Figure 1. Ultimately, 97 studies were included in this review. Table 1 summarizes the included studies with the exception of case reports. No potential biases were identified in the individual studies that met the inclusion criteria as all studies were evaluated based on reproducibility, methodological quality, and credibility.

This systematic review of the literature included 9098 patients living in different geographic regions, plus a study in Taiwan that included 1,051,148 patients with lung cancer from longitudinal health insurance databases (Table 1). Fifty-six studies are case report studies mainly associated with recurrent respiratory papillomatosis that progressed to lung cancer [17,18,19,20,21,22,23,24,25,26,27,28,29,30,31]. Only 18 publications are case–control studies in which normal lung tissue was used as a control [18,32,33,34,35,36,37,38,39,40,41,42,43,44,45,46,47,48].

**Table 1 cancers-16-03325-t001:** A summary of the included studies (with the exception of case reports).

				Histology Type												HPV Genotype		
Author	Year	Geographic Location	Patients (n)	ADC (n)	SCC (n)	SCLC (n)	LCC (n)	Other (n)	Gender Women (n)	Mean Age (Years)	Smokers (Including Ex-Smokers) (n)	Non-Smokers (n)	Sample for HPV Detection	HPV Detection	HPV Positive (%)	HPV16	HPV18	Others
Green, M. [49]	1981	USA	32	3	25	3		4	-	-	-	-	-	Southern blot	13.30%			
Stremlau, A. [50]	1985	Germany, Europe	24	-	-	-	-	-	-	-	-	-	-	Southern blot	4.10%			
Ogura, J. [51]	1993	Japan, Asia	29	-	-	-	-	-	-	-	-	-	fresh-frozen lung tumor tissue	PCR and Southern blot	10.30%			
Hirayasu, T. [51]	1996	Japan, Asia	43	-	43	-	-	-	4	68.3	-	-	FFPE	PCR for E6 and E6 and in situ hybridization	53%			
Welt, A. [52]	1997	Germany, Europe	38	-	32	6	-	-	-	-	-	-	FFPE	PCR and in situ hybridization	0			
Papadopoulou, K. [53]	1998	Greece, Europe	52	-	52	-	-	-	-	-	-	-	fresh-frozen lung tumor tissue	PCR and Southern blot	69%			
Bohlmeyer, T. [54]	1998	Colorado, USA	34	-	34	-	-	-	5	60.8	34	-	FFPE	PCR HPV L1 and Southern blot	5.88%			
Clavel, C. [55]	2000	France, Europe	185	60	101	-	4	12	30	66	185	-	FFPE	Hybrid Capture II	2.70%			
Miasko, A. [56]	2001	Poland, Europe	40	13	22	-	5	-	-	-	-	-	-	-	10%			
Miyagi, J. [57]	2001	Japan, Asia	121	62	59	-	-	-	-	-	-	-	FFPE	Southern blot and non-isotopic in situ hybridization (NISH)	33.80%			
Cheng. Y-W. [39]	2001	Taiwan, Asia	141	83	58	-	-	-	45	63	63	78	FFPE	nested PCR and in situ hybridization	54.60%	HPV 16/18		
Zafer, E. [58]	2004	Turkey, West Asia	40	40	-	-	-	-	40	-	-	-	fresh-frozen lung tumor tissue	PCR-RFLP	5%			
Cheng, Y.W. [32]	2004	Taiwan, Asia	141	83	58	-	-	-	45	-	64	77	FFPE	nested PCR consensus primers MY09 and MY11 followed by amplification with L1 primers and immunohistochemistry	35.70%	HPV 6 and HPV 11		
Coissard, C. [59]	2005	France, Europe	218	80	126	-	11	1	29	61.2	218	0	fresh-frozen lung tumor tissue	PCR followed by blot assay and viral load by real-time PCR	2%	x		
Jain, N. [33]	2005	India, Asia	40	9	21	9	1	-	5	-	32	8	biopsy tissue	PCR L1 region	5%	HPV 18		
Lin, T.S. [60]	2005	Taiwan, Asia	57	29	28	-	-	-	32	-	-	57	FFPE	nested PCR consensus primers MY09 and MY11 followed by amplification with L1 primers	50.90%	HPV16, HP18		
Ciotti, M. [61]	2006	Italy, Europe	38	15	16	-	4	1	5	65	34	4	21 FFPE and 17 fresh surgical specimens	PCR with primers for E6/E7, sequencing and RT-PCR for expression	21%	15.70%		
Castillo, A. [62]	2006	Latin America (Colombia, Mexico, Peru)	34	13	14	9	-	-	14	-	-	-	FFPE	PCR using HPV GP5+/GP6 combined with Southern blot hybridization	28%	20.58%	0.0588	0.0882
Wang, J. [63]	2006	Taiwan, Asia	155	89	64	-	-	-	59	-	88	65	FFPE	nested PCR consensus primers MY09 and MY11 followed by amplification with L1 primers	59.90%			
Carlson, J.W. [64]	2007	Boston, USA	22	-	22	-	-	-	-	-	-	-	FFPE	PCR and in situ hybridization	0			
Giuliani. L. [65]	2007	Italy, Europe	78	27	33	-	6	12	13	-	67	11	fresh-frozen lung tumor tissue	PCR E6/E7 oncogenes, RT-PCR for HPV transcripts	11.50%	7.69%		
Nadji, S. [35]	2007	Iran, West Asia	141	16	104	18	3	-	27	66.3	101	28	FFPE	nested PCR GP5+/GP6	6.00%			
Aguayo, F. [66]	2007	Chile, Latin America	69	32	37	-	-	-	19	-	49	12	FFPE	PCR and Southern blot	29%	15.94%		
Cheng, Y.W. [67]	2007	Taiwan, Asia	122	-	-	-	-	-	57	-	-	-	FFPE	nested PCR consensus primers MY09 and MY11 followed by amplification with L1 primers and immunohistochemistry	45%			
Park, M. S. [68]	2007	Korea, Asia	112	53	-	-	-	-	22	59	92	20	tumor tissue	PCR		10.70%	0.107	
Wang, Y. [40]	2008	China, Asia	313 (NSCLC)	-	-	-	-	-	76	54.8	208	105	fresh-frozen lung tumor tissue	PCR and non-isotopic in situ hybridization	44.10%			
Koshiol, J. (EAGLE study) [44]	2010	Italy, Europe	450	246	137	-	-	-	92	67.6	418	30	FFPE	PCR for HPV 16 and HPV 18	0			
Iwakawa, R. [69]	2010	Japan, Asia	275	275	-	-	-	-	171	60	140	108	FFPE	multiplex and nested PCR with HPV specific primers	0			
Aguayo, F. [70]	2010	Asia (China, Pakistan, Papua New Guinea)	60	38	18	3	-	-	17	-	-	-	FFPE	PCR qRT-PCR (viral load)	13%			
Joh. J. [71]	2010	USA	30	18	7	-	-	5	16	65	22	6	fresh-frozen lung tumor tissue	PCR for HPV L1	16.70%			
Zhang, J. [42]	2010	China, Asia	104	-	-	-	-	-	-	-	-	-	-	analysis (PCR, RT-PCR, or sequencing) of sequence variants of E6 and E7 oncogenes and L1 of HPV16	17.30%	HPV16		
Wang, Y.-H. [41]	2010	China, Asia	45	-	45	-	-	-	-	-	-	-	fresh-frozen lung tumor tissue	PCR	42.20%			
Baba, M. [72]	2010	Japan, Asia	57	30	27	-	-	-	18	66.5	40	17	FFPE	PCR with SPF10 primers and INNO-LiPA HPV genotyping assay		34%		
Halimi, M. [43]	2011	Iran, West Asia	30	-	30	-	-	-	17	66	-	-	-	PCR for MY11 and MY09 primers	10%			
Goto, A. [73]	2011	Asia	304	176	128	-	-	-	-	-	-	-	FFPE	PCR using HPV GP5+/GP6; in situ hybridization	15	majority was positive for HPV 16/18		
Galvan, A. [74]	2012	Milan and London, Europe	100	72	20	-	-	-	-	65	81	19	fresh-frozen lung tumor tissue	HPV DNA detection, followed by PCR amplification of L1, followed by incubation with specific probles	0			
Gatta, L. [75]	2012	Italy, Europe	50	-	50	-	-	-	2	68	-	-	FFPE	PCR of L1 region and E6/E7 region of high-risk viral genotype	4.00%			
Van Boerdonk [76]	2013	The Netherlands, Europe	223	-	-	-	-	-	78	65	-	-	archival tumor tissue specimens	PCR; HPVE/RT-PCR	1.34%			
Yanagawa, N. [9]	2013	Canada, North America	336	204	132	-	-	-	145	67.7	-	-	FFPE surgical samples	in situ hybridization (INFORM HPV III Family 6 and 16 probes); PCR	1.50%	1.49%		
Márquez-Medina, D. [77]	2013	Spain, Europe	40 (NSCLC)	-	-	-	-	-	20	-	30	20	12 biopsies in paraffin, 14 cell buttons obtained by fine-needle aspiration and preserved in paraffin, 14 cytology samples by fine-needle aspiration	linear array HPV genotyping test	2.50%			
Jafari, H. [78]	2013	Irão, Asia	50	-	50	-	-	-	13	65.4	-	-	FFPE	nested multiplex PCR and direct DNA sequencing	0.18	2%	0.16	
Badillo-Almaraz, I. [79]	2013	Mexico, Central America	39	18	21	-	-	-	14	55	27	12	-	PCR using HPV GP5+/GP6+, genotyped for HPV16/18, in situ hybridization	41.02%	38.46%	2.34%	
Sagerup, C. [80]	2014	Norway, Europe	334	111	28	-	-	195	168	66.1	307	27	fresh-frozen lung tumor tissue	nested multiplex PCR assay	3.90%			
Fan. X. [81]	2014	Shanghai, China, Asia	262	128	134	-	-	-	64	-	52	56	FFPE	reverse dot blot and PCR for presence of HPV DNA	8.40%	5.34%	2%	
Anantharaman, D. [45]	2014	Europe	334	-	-	-	-	-	84	58	233	101	FFPE and snap-frozen tumor tissues	PCR-MPG assay	9.70%	6.60%		2.40%
Sarchianaki, E. [46]	2014	Greece, Europe	100	50	39	-	-	67	9	-	93	7	FFPE	RT-PCR using GP5+/GP6+, genotyping	19%			
Hartley, C. P. [37]	2015	New Hampshire, USA	20	-	-	20	-	-	54	66.3	-	-	FFPE surgical/biopsy samples	Roche linear array HPV genotyping test (Roche) and Cobas HPV genotyping	0			
Colombara, D.V. [36]	2015	USA, North America	200	146	-	44	-	10	82	61.6	151	-	blood–liquid bead microarray antibody-Bio-Plex 200	-	no association with seropositivity			
Yu, Y. [47]	2015	China, Asia	194	56	88	36	-	-	57	51. 60	134	60	FFPE	qPCR (HPVL1, HPV16, HPV18)		37.22%	0.3111	
Isa, S. [82]	2015	Osaka, Japan, Asia	96	85	-	-	-	-	82	67	0	100	FFPE	PCR-based microarray (E2, E6, E7, L1, and L2 genes)				
Robinson [83]	2016	Florida, USA	70	27	33	-	-	10	-	-	-	-	snap-frozen tumor	microarray, oncovirus screening panel, genotyping by PCR	0.20%			
Kawaguchi, T. [84]	2016	Soaka, Japan, Asia	876 (NSCLC)	-	-	-	-	-	457	70	441	435	formalin-fixed paraffin-embedded surgical samples	PCR-based microarray system	0.30%			
Lee, J.E. [3]	2016	Korea, Asia	233	99	134	-	-	-	-	-	-	-	-	-	0.45%	episomal or integrate high-risk HPV analysis is negative in all 233 cases		
Lin, F.C.F. [85]	2016	Taiwan, Asia	1,051,148	-	-	-	-	-		48	-	-	longitudinal health insurance databases	-	2.30%			
Ilahi, N. [86]	2016	Pakistan, Asia	9	3	5	-	-	-	-	-	-	-	FFPE	PCR for HPV 16 and HPV 18	11.10%	HPV 16		
Wu, Y.L. [87]	2016	Taiwan, Asia	319 (NSCLC)	-	-	-	-	-	101	-	145	195	surgery samples	-	28.50%			
Lu, Y. [48]	2016	China, Asia	72	24	48	-	-	-	20	-	46	28	frozen tissues	PCR HPV16E6 and HPV18E7	45.83%	19.44%	0.2637	
Argyri, E. [7]	2017	Athens, Greece, Europe	67	31	20	12	-	-	9	67.6	65	1	biopsies collected during bronchoscopy (conserved in ThinPrep)	PapilloCheck^®^ HPV genotyping assay	3.00%	x		
Oliveira, T. [88]	2018	Brazil, Latin America	63	-	-	-	-	-	-	59	31	22	FFPE	PCR; HPV18/18 E6 and E7 by immunohistochemical stain	52.38%	41.50%	0.1088	
Jaworek, H. [89]	2019	Czech, Europe	80	27	42	-	9	2	26	-	-	-	FFPE and fresh-frozen samples	qPCR	0	x	x	x
Silva, E. M. [90]	2019	Brazil, Latin America	62	21	41	-	-	-	-	-	41	15	FFPE	multiplex PCR and HPV16 specific real-time PCR	0			
He. F. [91]	2019	China, Asia	140	88	38	-	-	-	53	-	54	-	tissue cryopreserved (−80 °C) within 10 min after surgery	Hybribio HPV Test kit (21 common types of HPV)	9.29%	2.86%	1.43%	3.57%
Wu, Y. [92]	2020	China, Asia	100	14	2	-	-	-	57	-	-	-	circulating HPV DNA	PCR and sequencing for confirmation	16%	HPV 16		
Hussen, B. M. [18]	2021	Tehran and Kermanshah, Iran, West Asia	109	35	57	17	-	-	28	57. 41	74	35	fresh tissue snap-frozen in liquid nitrogen	INNO-LiPA HPV Genotyping Extra II on the Tendigo^®^ platform	51.40%	41.10%		
Rojas L. [2]	2022	Colombia, Latin America	133	133	-	-	-	-	66	67	26	84	FFPE adenocarcinoma	INNO-LiPA HPV Genotyping Extra II on the Tendigo^®^ platform	25.80%			
Chen, M.J. [93]	2022	Taiwan, Asia	223	116	107	-	-	-	73		102	121	FFPE	HPV16/18 E6 oncoprotein expression	26.90%			
Tsyganov, M. [94]	2023	Russia, Asia/Europe	274 (NSCLC)	-	-	-	-	-	31	60.8	-	-	fresh-frozen lung tumor tissue	commercially available kits for PCR	12.70%			
Sun, W. [95]	2023	China, Asia	102	102	-	-	-	-	62	63	28	74	FFPE	flow fluorescence	28.43%			
Harabajsa S. [96]	2023	Zagreb, Croatia, Europe	67	67	-	-	-	-	-	69	-	-	lung adenocarcinoma cytological smears	PCR	43%			

ADC, adenocarcinoma; SCC, squamous cell carcinoma; SCLC, small-cell lung cancer; LCC, large-cell cancer; NSCLC, non-small-cell lung cancer; HPV, human-papillomavirus; FFPE, formalin-fixed paraffin-embedded; PCR, polymerase chain reaction.

### 3.2. Diagnostic Criteria

The criteria used for the diagnosis of lung cancer are not explained in the studies included in this research. The main samples used for HPV detection were formalin-fixed paraffin-embedded (FFPE) tumor tissue and fresh-frozen lung cancer tissue (Table 1). This choice of sampling could justify some discrepancies found in the results, even among the same geographic localization. HPV detection was more frequent in the fresh-frozen tissues than in FFPE specimens [40,61]. Furthermore, the diagnostic criteria used for the identification of HPV in lung cancer varied in the different studies. Southern blotting and PCR followed by Southern blotting were used in the earlier studies spanning from 1981 until 2000 [49,50,51,52,53,54,97]. In contrast, more recent studies employed multiplex or commercial kits for PCR, followed by sequencing and HPV16/18 E6 oncoprotein expression [2,18,90,92,93,94,95,96]. Of note, technical issues may affect HPV positivity rates and thus influence the detection results obtained. In fact, the detection of HPV DNA through PCR must be carefully monitored and analyzed. Since PCR is an extremely sensitive technique, it can detect HPV that is not biologically significant, such as episomal or transient forms [98]. Additionally, the use of specific primers designed with higher sensitivity and specificity for specific HPV genome regions (such as E6 and E7) allow for a more accurate diagnosis in comparison to primers used for the detection of several HPV types [69,88].

### 3.3. Epidemiology and Etiology

This systematic review comprised 5085 patients from Asia, 2792 patients from Europe, and 696 patients from America. The percentage of HPV infections in the studies included in this systematic review of the literature varied from 0% to 69% without exhibiting any discernible increasing or decreasing trend over the years (Table 1 and Table 2).

Data regarding the patients’ mean age was not available in all studies. Only 35 studies (Table 1) reported the mean age, which was found to be 63 years old across all patients included. Gender information was reported in 52 studies, revealing that approximately 30% were female. In fact, although most studies did not find a correlation between gender and HPV infection in lung cancer, some additional studies conducted in Taiwanese populations showed that HPV16/18 was mainly detected in female non-smokers [39,99,100]. These observations imply that further investigations are warranted to explore potential gender-specific differences in HPV prevalence and its clinical implications. In addition, the histological type was reported in 47 studies, with 3147 (51.5%) being adenocarcinomas, 2430 (39.7%) being squamous cell carcinomas (SCCs), 177 (2.9%) being small-cell carcinoma (SCLC), and 43 (0.75%) being large-cell carcinoma, and 319 (5.2%) cases were classified as other histological types. Moreover, smoking habits were reported in 42 of the included studies: 4016 (66.4%) patients were smokers (including past smokers), and 2032 were non-smokers (33.6%).

In a multi-institutional study involving Asian populations, inclusive of HPV-endemic areas, rates of HPV positivity of 6.3% in lung SCC and 7.0% in adenocarcinoma were reported, which are relatively low rates in Asian populations [73]. In Japan, Kawaguchi et al. conducted a prospective study that showed little evidence of HPV in early-stage NSCLC [84]. Similarly, another group in Japan suggested that HPV does not play a major role as the driving oncogenic event in lung cancer among non-smokers [82]. This result was also supported by Iwakawa’s study, which used multiplex and nested PCR, indicating a limited role of HPV in lung adenocarcinoma etiology [69]. Another Japanese study demonstrated that although HPV16 may be more strongly associated with adenocarcinoma than squamous cell carcinoma, particularly in gefitinib-responsive adenocarcinoma, the low viral load makes the interpretation of these findings difficult [72]. Notably, a study with a relevant number of patients in Shanghai, China (1262 cases) showed an overall HPV infection rate of 8.40% with a higher prevalence in patients with SCC compared to ADC [81]. In the SCC group, the infection rate was significantly higher in females than in males and in patients older than 60 years [81]. Of note, the disparity found could be related to the sample mode (fresh-frozen versus FFPE) and the detection method.

In North America, Hartly et al. could not find the presence of HPV in any of the 20 samples of SCLC, suggesting no association between high-risk HPV infection and the pathogenesis of SCLC in this region [37]. Similarly, in a cohort of 336 North American Canadian patients with NSCLC, the overall frequency of HPV infection was 1.5%; only five patients with SCC were non-smokers and genotyped as HPV16 [9]. Notably, these five patients had a previous history of potentially related head and neck (n = 2) and cervical squamous cell carcinoma (n = 3) [9]. Of note, in this study, HPV was detected by PCR and in situ hybridization, and a concordance of 100% was found [9]. In Mexico, HPV was detected in 16/38 samples of lung cancer without a correlation with gender, smoking status, or histology [79]. In Chile, HPV was present in 29% of lung cancers, with HPV16 being the most frequent genotype and significant differences being observed between HPV positivity and histological characteristics, particularly in SCC tumors. Similarly, a study involving three Latin American countries detected high-risk HPV in 28% of lung carcinomas [62]. In Brazil, in a study performed by Silva et al., no HPV DNA was detected among 62 patients with NSCLC [90], while in 2018, Oliveira et al. detected HPV16 and HPV18 in 33 out of 63 patients, which was predominantly distributed among different subtypes: 39.39% in SCC, 33.33% in adenocarcinomas, and 18.18% in SCC. Furthermore, the presence of E6 and E7 proteins from HPV16 and HPV18 was detected by immunohistochemistry in 28 out of the same 33 HPV positive samples (84.85%) and 25 out of the same 33 HPV positive samples (75.76%), respectively [88].

In Europe, a Dutch population study involving 223 patients detected HPV in only three patients, all belonging to a group with equivocal lung cancer history, possibly indicating metastasis rather than primary lung tumors [76]. A retrospective analysis of 40 NSCLC samples in Europe revealed HPV in one patient (2.5%) with no further associations due to the limited sample size [77]. In a Czech population including 80 patients with NCSCL, HPV (16, 18, 31, or 56) was not detected in any of the samples by qPCR [89]. In Greece, HPV was detected in 19% of patients with NSCLC, with the HPV16 genotype being the most prevalent genotype, but no further associations were observed with demographic, histology, smoking habits, or lung functional parameters [46]. Similarly, in Norway, HPV genotypes were detected in 3.9% of 334 lung cancer samples, with no correlation found with histology, smoking status, or EGFR/KRAS mutations [80]. Notably, discrepancies in HPV detection were observed in studies from Italy, with one reporting a high HPV detection rate in patients with lung cancer and others finding no HPV DNA in tumor specimens nor in normal lung tissue [74,75]. These are clear examples of the discrepancies found in studies, even within the same geographic localization. A representative study of the Western population, which included Italian patients, showed that the presence of HPV DNA was not associated with the development of lung cancer [44]. However, a study in 38 samples of NSCLC demonstrated that E6 and E7 oncogenes for specific HPV types were present in 21% of the patients solely in tumor tissue and not in the surrounding normal tissue [61]. Another study also including Italian patients with lung cancer revealed that 11.5% of the evaluated tumors were positive for HPV, and almost all, except one, were also positive for E6 and E7 transcripts [34]. Notably, HPV detection was more frequent in fresh-frozen tissues than in FFPE specimens and, among the positive tumors for HPV, all except one were positive for E6 and E7 transcripts [61]. In 2014, a larger study in Europe evaluated the HPV genotype in retrospective and prospective cohorts. Almost 10% of the lung tumors were positive for HPV DNA; however, none expressed the viral oncogenes [45,75]. 

Case–control studies focused on the potential role of HPV and lung cancer development have also been reported. Overall, higher HPV infection rates were detected in lung tumor tissues when compared to non-malignant tissue controls [18,32,33,34,35,36,37,38,39,40,41,42,43,44,45,46,47,48]. In a recent study developed in Iraq, 109 lung cancer tissues and 52 controls were evaluated, and the percentages of HPV genome detection were 51.4% and 23.1%, respectively [18]. Additionally, in Taiwan, a significantly higher prevalence of HPV16/18 DNA, particularly in non-smoking female patients, was found in lung tumor cells (54.6%) when compared to controls without cancer (26.7%) [39]. Also, a significantly higher HPV6 infection rate, particularly in male patients, was reported in lung tumors (28.4%) in comparison to the non-malignant controls (1.7%) [32]. Similarly, a study from Iran showed that 33 of 129 lung tumors had HPV DNA compared with 8 of 90 control subjects without cancer (25.6% vs. 9.0%; *p* = 0.002) [35]. Moreover, in a study developed in China, the detection rate of HPV DNA observed in non-small-cell lung cancer (NSCLC) pathologic tissue samples (45.83%) was significantly higher when compared to benign lung lesions used as a control group (3.7%) [48]. Recently, the expression levels of E6 mRNA and E7 mRNA in HPV16 were evaluated by qRT-PCR in 310 patients with lung cancer and compared with benign lung diseases. Interestingly, the mRNA levels of E6 and E7 oncoproteins were significantly higher in the group of patients with lung cancer when compared to the group with benign lung diseases [38]. Nonetheless, some case–control studies do not show evidence of a strong association between HPV infection and lung cancer development [44,45]. Of note, a case of HPV infection in an extremely rare lung cancer tumor (mixed squamous cell and glandular papilloma of the lung) was recently described for the first time [101]. Indeed, solitary pulmonary papillomas (SPPs) are rare neoplasms, accounting for less than 0.5% of all lung cancers. Notably, Hung et al. detected in 2019, for the first time, a case of mixed papilloma (one of the three types of SPPs) with PCR-confirmed infections of the HPV genotypes 16, 35, and 51 in an 18-year-old non-smoking male. These results may shed some light on the etiology and pathogenesis of HPV in young non-smoking patients and suggest that HPV may play a pathogenetic role in mixed papilloma [101].

### 3.4. Prognosis

Besides these studies attempting to correlate the presence of HPV with lung cancer, some research studies have explored the relationship between HPV oncoprotein expression and clinical outcomes in patients with primary lung cancer [102]. However, few studies have considered HPV infection as a prognostic factor in lung cancer [103]. Evidence from a recent study suggests that HPV 16E6/18E6 oncoproteins and epidermal growth factor receptor (EGFR) expression serve as good prognostic factors in patients with lung adenocarcinoma [103]. In this study, transfected HPV 16E5/16E6/16E7 NCI H292 lung mucoepidermal carcinoma cells were used to investigate the mechanism of HPV oncoproteins interfering with EGFR nuclear trafficking related to a better response to cisplatin. Interestingly, a significantly higher phosphorylated nuclear EGFR expression upon epidermal growth factor stimulus and better responses to cisplatin in transfected HPV 16E5/16E6/16E7 NCI H292 cells and xenograft animal models were observed [103]. Furthermore, the clinical effects of the combination of HPV 16E6/18E6 expression and EGFR expressions were also analyzed in 173 patients with lung adenocarcinoma, and it was found that patients with lung adenocarcinoma with E6+tEGFR+ had the longest survival time, particularly patients with an older age, no brain metastasis, smoking history, and a wild-type EGFR status [103]. Importantly, this study suggests that the expression of HPV 16E6/18E6 and EGFR may serve as a predictive biomarker of survival in patients with lung adenocarcinoma [103].

Chen et al., in 2012, demonstrated that HPV16/18 E6, but not HPV16/18 DNA, was inversely associated with p53 expression in lung tumors [102]. Thus, HPV 16/18 E6 protein involvement in the p53 inactivation that contributes to HPV-infected lung tumorigenesis is associated with p53 codon 72 genotypes [102]. In a study population of Taiwanese patients with lung cancer, a high frequency of TIMP-2 loss by LOH and/or promotor hypermethylation in E6-positive lung tumors was found compared with E6-negative lung tumors [104]. The other carcinogenic effect of HPV-16 oncoproteins in NSCLC is the promotion of tumor angiogenesis both in vitro and in vivo, and correspondingly, the enhanced expression of HIF-1α and VEGF [105]. HPV status could be an important biomarker and parameter in patients with NSCSL [102]. In tumors with HPV16/18E6 oncoprotein expression or mutant EGFR, PD-L1 expression is induced, promoting tumor growth and invasiveness in NSCLC [93]. This was shown in 223 surgically resected patients with NSCLC in which 60 tumors were positive for HPV16/18 E6 oncoprotein expression [93]. In another Asian study, the expression of HPV 16 E6/E7 mRNA found in 44 cases of small-cell lung cancer was significantly higher than that in the benign cell group [38]. Furthermore, in a recently published study including Russian patients with lung cancer, HPV was identified in 12.7% of patients, with a predominance of HPV16 and HPV18 [94]. However, among these patients, 74.3% showed a clinically insignificant viral load [94]. Importantly, regarding the prediction of the outcome, the presence of HPV was not a factor that significantly influenced the risk of tumor metastasis [94].

## 4. Discussion

Lung cancer is one of the most diagnosed cancers worldwide, and its high mortality rate underscores the imperative to explore a new facet of tumor biology [18]. Moreover, the recent increase in lung cancer diagnoses among non-smokers suggests the involvement of additional factors, such as viral infections, in lung cancer development [15,18].

HPV infections have been implicated in lung carcinogenesis, although causal associations remain uncertain [45,96]. Over the past four decades, there has been a significant increase in reports suggesting an association between HPV and lung cancer, namely HPV16, 18, and 56 [15,106]. However, evidence has not fully addressed this relationship due to inherent limitations, inconsistencies, methodological variations, gaps in causality, evolving research landscapes, and potential biases. Addressing these limitations and identifying areas requiring further investigation are crucial for providing a comprehensive and critical assessment of the current understanding of HPV infection as a potential factor in lung cancer development.

The results regarding the detection of HPV in lung tumors, particularly concerning the choice of sampling method, sample preservation quality, and diagnostic criteria employed, necessitate careful consideration and discussion. Variability introduced by the utilization of FFPE tumor tissue and fresh-frozen lung cancer tissue as the primary sampling methods for HPV detection poses challenges, with fresh-frozen tissue showing higher HPV detection rates compared to FFPE specimens. While FFPE tissue specimens can provide HPV nuclei acid evaluation, extensive DNA damage due to cross-linking and fragmentation results in poor yield [40,61]. Additionally, significant variability in diagnostic criteria across studies complicates the comparison of results and may contribute to inconsistencies in the HPV detection rate.

The detection of HPV DNA typically involves PCR and in situ hybridization (ISH) signal amplification in target tissues. ISH offers the advantage of direct visualization of HPV in either episomal form or integrated forms within the nuclei of tumor cells in tissues [107], while PCR, although extremely sensitive, may detect HPV that is not biologically significant, such as episomal or transient forms [98]. Furthermore, the choice of genome region used for HPV genotype detection can impact the results, with specific primers designed for E6 or E7 from specific HPV exhibiting higher sensitivity and specificity compared to consensus primers for several HPV types, such as those from the L1 region [69,88]. This could result in a higher prevalence of HPV in lung cancers analyzed by specific primers for each HPV [108]. To confirm that a tumor is caused by HPV, detecting the mRNA transcripts of HPV E6 and E7 viral oncogenes is necessary [3].

In addition to methodologic issues, the differences found among studies could be related to high heterogeneity among study participants such as geographical variations in epidemiological risk factors [90,96]. Consequently, an assertive understanding of the real significance of HPV in lung cancer is currently questionable. Moreover, geographic differences in HPV prevalence in lung tumor tissue may be associated with variations in smoking habits, sexual behaviors, or other factors related to environmental exposures, culture, topography, or genetics [44].

Histologically, adenocarcinoma and squamous cell carcinoma were the most common types of lung cancer observed among the patients. This distribution aligns with the known histopathological subtypes of lung cancer [109]. Few studies reported the percentage of HPV presence among histological types. Regarding lung adenocarcinoma, the etiology and pathogenesis are still controversial. One study reported that 43% of adenocarcinoma samples were positive for HPV infections [96]. Other studies reported 16% in a European population and 37% in an Asian population [96]. In a meta-analysis, 22.4% of lung cancer tissues, particularly adenocarcinoma and squamous cell carcinoma, had HPV, predominantly types 16 and 18 [108]. These numbers highlight the need for targeted studies focusing on these specific subtypes to further elucidate the role of HPV in their pathogenesis.

The association with oncogenic driver mutations and environmental factors remains incompletely understood [84]. Although HPV16 was the most frequent genotype found in lung cancer, its correlation with clinicopathological parameters such age, sex, smoking status, histological type, tumor stage, and grade was not consistently observed [88,91]. Furthermore, whether HPV infection is a risk factor for lung cancer in smoking patients is controversial. Smoking status was reported in a substantial proportion of patients, with approximately two-thirds being smokers or past smokers. However, the results were not consistent. In fact, while some studies have reported a higher HPV infection rate in smoking patients, others did not find a significant difference [47,91,110]. The interaction between tobacco and HPV infection for lung cancer could be important, with the squamous–columnar junction, a preferred entry site for HPV, frequently presenting multifocal metaplastic areas along the bronchial tree from which squamous bronchogenic carcinomas may arise [7,111]. Cigarette smoke may interact with HPV E6 and E7, activating the HPV16 p97 promotor in lung epithelial cells [7].

Cases of recurrent respiratory papillomatosis, caused by HPV, which can undergo malignant transformation to the lung, highlight the clinical implications of HPV-related lung diseases. Notably, pulmonary involvement is associated with worse clinical outcomes, with HPV11 specifically linked to an increased risk of malignancy in these patients [17,112,113]. HPV11 is considered a low-risk HPV; however, it is associated with benign warts that could occur in anogenital regions and could be transmitted vertically from mother to child around the time of conception, during pregnancy, or at delivery or immediately thereafter [114,115]. The risk of developing recurrent respiratory papillomatosis seems to be 231 times higher in children born to mothers with an active HPV infection than in children born to unaffected mothers [116].

Regarding prognosis, limited studies have specifically investigated HPV infection as a prognostic factor in lung cancer. Although there is still some controversy about the role of HPV in lung carcinogenesis, there are published studies in the literature that support and/or reinforce the prognostic value of HPV infection in lung cancer development. Previous studies have demonstrated that HPV infection, especially HPV16 and 18, is associated with a higher risk of lung cancer development [117,118] despite substantial variation in the viral prevalence between geographic regions [117]. Recently, in a study developed in Brazil, the presence and activity of HPV was evaluated in patients with lung cancer. A high prevalence of HPV, particularly HPV16 and 18, was found in lung cancer biopsies, and it was shown that the virus was active and expressing E6 and E7 oncoproteins in lung tumor cells. These results suggest transcriptional activity in tumor cells, implicating HPV in lung carcinogenesis [2,88,117]. Notably, a recent study by Hussen et al. demonstrated that HPV infection and its interaction with cellular genes and miRNAs promote epithelial–mesenchymal transition (EMT), which is involved in lung cancer development [18]. Of note, these observations emphasize the importance of the physical status of the HPV genome as a marker for tumor progression (Figure 2). Also, it was recently shown that HPV infection is of prognostic significance in patients with lung adenocarcinoma treated with immunotherapy [2]. Indeed, it was observed that patients with HPV+ lung adenocarcinoma had a significant benefit in the overall response and survival outcomes when compared to patients without HPV [2]. Additionally, it was recently suggested that HPV 16E6/18E6 oncoproteins and epidermal growth factor receptor (EGFR) expression may serve as good prognostic factors in patients with lung adenocarcinoma [103]. Nevertheless, cumulative prognostic evidence is still insufficient, and further studies are required.

The transmission of HPV to the lungs is another issue of interest, with HPV DNA and mRNA being detected in the peripheral blood of patients with cervical cancer [85]. The spread of HPV from superficial sites via blood to the lungs, particularly peripheral sites prone to adenocarcinoma, may explain why female non-smokers infected with HPV are more likely to develop adenocarcinoma [85] (Figure 3).

This systematic review showed that several types of HPVs, particularly HPV16 and 18, have been found in the lung tissue of patients with lung cancer. This result was further confirmed by a recent mendelian randomization study in which HPV18 E7 protein exposure was associated with a higher risk of the development of lung cancer [95]. However, while this systematic review provides valuable insights into the potential association between HPV infection and lung cancer, several limitations should be acknowledged, which may affect the interpretation and generalization of the findings: the included studies varied significantly in terms of study design, sample size, patient demographics, and methodology for HPV detection. However, individually, the quality of the studies did not impact the results of this study.

This heterogeneity makes it challenging to directly compare results across studies and draw definitive conclusions. Furthermore, inconsistencies in diagnostic criteria, sampling methods, and HPV detection techniques introduce additional variability into the findings. The prevalence of HPV infection in lung cancer tissues may vary based on geographic location, environmental exposures, cultural factors, and patient demographics. Variations in smoking habits, sexual behaviors, and other environmental factors among different populations may confound the association between HPV infection and lung cancer risk. Most included studies are cross-sectional or retrospective in nature, limiting the ability to establish temporality and causality in the observed associations. Longitudinal studies with a comprehensive follow-up and mechanistic investigations are needed to elucidate the dynamic interplay between HPV infection and lung cancer development over time. Therefore, since the role of HPV infection in lung cancer development is still controversial, the use of recommended methods for HPV detection is crucial when trying to establish a link between viral infection and lung carcinogenesis (Figure 4). Nevertheless, further research studies are needed to clarify the role of HPV in lung cancer development.

In addition, an important consideration is vaccination against HPV. Given that HPV vaccines became available in 2006 (quadrivalent) and the nonvalent vaccine only became available in 2016, there is a significant number of patients already infected with high-risk HPV who may be at risk of developing lung cancer. Therefore, further studies are needed to clarify the potential role of HPV vaccination in preventing lung cancer [119].

## 5. Conclusions

In conclusion, this systematic review provides valuable insights into the epidemiology and clinical characteristics of HPV-associated lung cancer. The increasing incidence of lung cancer, particularly among non-smokers, underscores the importance of exploring novel factors contributing to its development, such as viral infections like HPV. While numerous studies have suggested an association between HPV and lung cancer over the past four decades, the evidence remains inconclusive due to various limitations and challenges. Despite these limitations, several intriguing associations have emerged, including the potential role of HPV, particularly HPV16/18, in lung adenocarcinoma and squamous cell carcinoma. However, the correlation with clinicopathological parameters and prognostic significance remains uncertain and warrants further investigation. Further research utilizing standardized methodologies and larger patient cohorts is needed to elucidate the role of HPV in lung carcinogenesis and its potential implications for diagnosis, treatment, and prognosis.

## Figures and Tables

**Figure 1 cancers-16-03325-f001:**
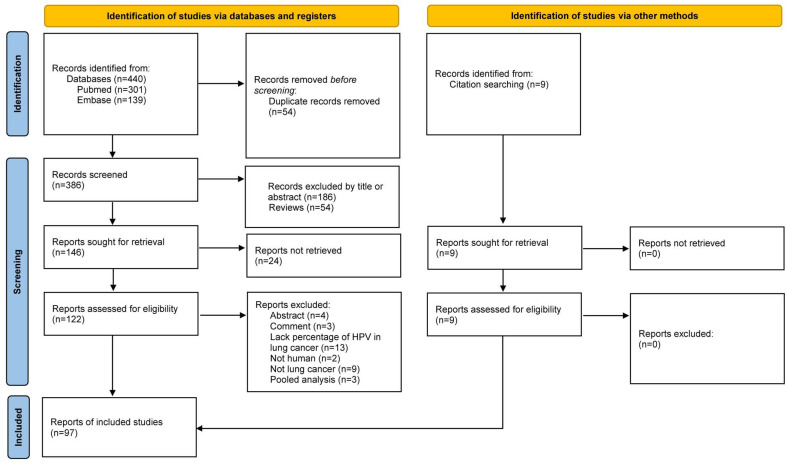
A PRISMA flow chart for the systematic review of the literature adapted from Page MJ et al. [16].

**Figure 2 cancers-16-03325-f002:**
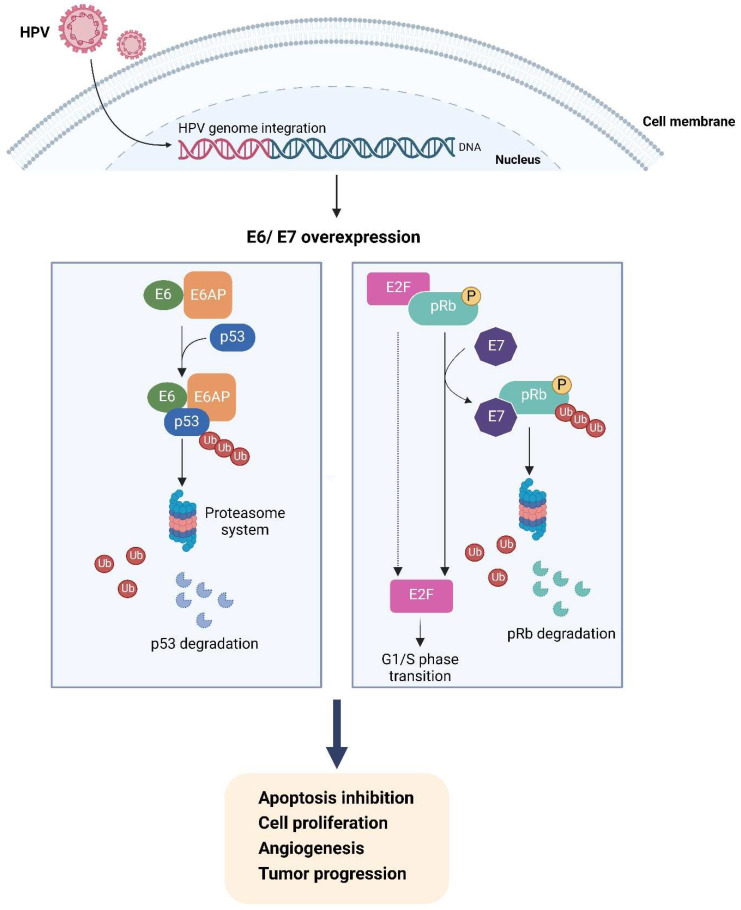
Possible molecular mechanisms of HPV pathogenesis of lung cancer [38,93,102,104]. Created with BioRender.com (https://www.biorender.com, accessed on 30 July 2024).

**Figure 3 cancers-16-03325-f003:**
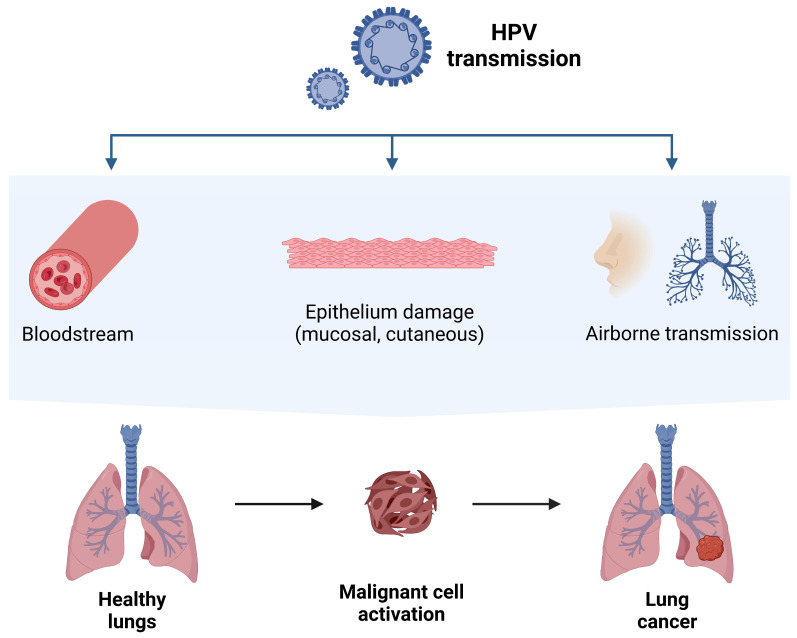
HPV transmission routes to the lungs. HPV DNA has been identified not only in human neoplastic lung cells, but also in serum, plasma, and peripheral blood mononuclear cells. Therefore, along with oral and airborne routes, the bloodstream itself could be one of the pathways of transmission from infected organs to the lungs [94,106,119]. The figure was created with BioRender.com (https://www.biorender.com, accessed on 26 September 2024).

**Figure 4 cancers-16-03325-f004:**
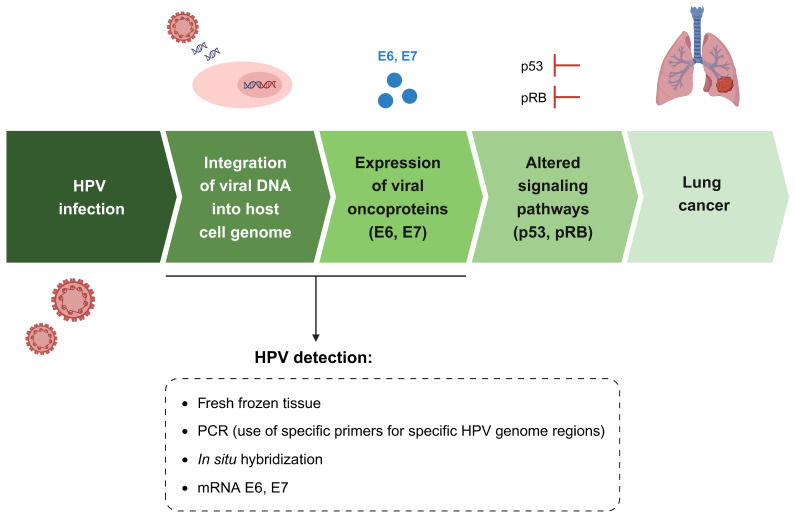
A schematic representation of HPV infection and lung cancer development. The recommended methods for HPV detection are crucial for an accurate diagnosis and include the preferable use of fresh-frozen tissue samples, the PCR technique with highly specific primers for specific HPV genome regions, in situ hybridization, and the evaluation of the expression of HPV E6/E7 oncogenes by means of mRNA detection. The figure was created with BioRender.com (https://www.biorender.com, accessed on 27 September 2024).

**Table 2 cancers-16-03325-t002:** Epidemiology of HPV infection in different geographic areas.

Geographic Localization	HPV Infection in Lung Cancer (Min-Max)	Type of HPV Detected *
America (14 studies)	0–52.38%	HPV16 (1.49–41.50%) HPV18 (2.34–5.88%)
Asia (35 studies)	0–59.90%	HPV16 (2–41.10%) HPV18 (1.43–31.11%)
Europe (20 studies)	0–69.00%	HPV16 (6.60–15.70%)

* Most of the studies did not report the type of HPV.

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
