# Peer review of "HPV and Lung Cancer: A Systematic Review"

_cancers, 2024, doi:10.3390/cancers16193325_

Round 1

Reviewer 1 Report

Comments and Suggestions for Authors

The authors attempt to summarize and review the available literature on HPV and lung cancer. I have several concerns that are detailed below.

My primary concern is that the review does not effectively summarize the studies included (other than including a large table with details of each study).  There is no real attempt to combine or synthesize the data from the available studies to attempt to draw any meaningful conclusions.  The authors primarily pick small facets from some of the studies and briefly detail them in somewhat haphazard fashion (primarily referring to Section 3.3 which I believe would require an extensive re-write).  

Admittedly, as the authors effectively point out, the studies are very heterogeneous in terms of patient population, data reported, how HPV status was determined, and study design which poses challenges in terms of drawing firm conclusions.

Some specific questions/concerns:

In the Introduction, it would be helpful to include how your study differs from the recent 2021 meta-analysis, e.g. how many studies were added compared to that review.  It appears this review include fewer patients than the one in 2021. 

On page 4, line 132 you detail that 18 of the studies were case control, yet there is no attempt in Section 3.3 to summarize this data to gain some insight on whether there is truly an increase in HPV in lung cancer relative to normal controls.  I would suggest adding such a summary.

On page 12, line 143 the sentence starting with "Positivity" does not make sense as written.

In section 3.2, again there is no attempt to combine or summarize the data.  For instance, is there any association between the type of HPV testing utilized and positivity rates?

In Table 2 on page 12, the column titled "Histology type" does not make sense to me.  Are these percentages the percentage of HPV+ tumors that were this histology.  If so, shouldn't it be a range (as in the first 2 columns)?  Also, in row 3 the percentages only add up to roughly 81%.  What were the other 19%?

On page 13, line 199, I believe the sentence starting with "Furthermore" should clarify that the 33 samples referred to are the 33 samples (I assume) that were mentioned as being positive for HPV detection in the prior sentence.

Page 14, lines 228 - 232, should be moved to Section 3.4 as it seems more relevant to the Prognosis section of the paper.

Page 14, lines 233-240, I am not sure including the RRP population adds much as this is really a unique entity with known HPV association

Page 14, lines 245-251, I think the Yang case could be omitted as it does not appear to have anything to do with SPPs (which is what is referenced at the beginning of the paragraph) and it also may be a case of metastatic cervical cancer.  In addition it states that the patient has the "same" HPV type.  Not sure what :"same" refers to as there is no mention of HPV 18 earlier in the paragraph.

Page 14, lines 257-261, this sentence is not clear as written and requires considerably more detail

Page 16, lines 322-326, this appears to be a statement that should be included in Section 3.3, not in the Discussion, as it is detailing results that have not been mentioned previously.  It would also need to be presented with the caveat that many of the studies did not include gender (as you detailed previously).

Page 16, lines 361-270, this paragraph leads with "Regarding prognosis..." ye the majority of the paragraph does not discuss prognosis. Suggest re-writing.

Page 18, lines 388-390, I disagree with the assertion that this review showed that "infection of HPV 16/18 was associated with the development of lung cancer." No compelling data was provided that supports that conclusion.

Comments on the Quality of English Language

The English requires extensive revision

Reviewer 2 Report

Comments and Suggestions for Authors

Dear authors,

Cancers-3159162

HPV and lung cancer: a systematic review by Sequeira et al summarizes the role of human papillomavirus (HPV) in lung cancer development. This systematic review aims to explore 20 the diagnostic criteria, epidemiology, aetiology, and prognosis of HPV infection in lung cancer. A total of 97 studies encompassing 9,098 patients worldwide, revealing varied HPV infection rates in lung cancer, ranging from 0% to 69%, were analyzed. While HPV16/18 was predominant in some regions, its association with lung cancer remained inconclusive due to conflicting findings. Some studies suggested a limited role of HPV in lung carcinogenesis, particularly in non- smokers. Despite inconclusive evidence, intriguing associations between HPV and lung adenocarcinoma and squamous cell carcinoma have emerged. Further research with standardized methodologies and larger cohorts is needed for clarity.  The review presents a nice summary; however; the study needs additional information to gain interest among wide cancer and non-cancer researches.

Major comment:

1.     The authors summarize the past research.  The HPV infection deregulates p53 and pRB pathway and this known for a quite long time. The authors could summarize how to resolve this contradiction in a cartoon model and conclude their hypothesis will enhance the quality of the manuscript.

2.     The manuscript needs s strong English edit.

Comments on the Quality of English Language

Dear editor,

Cancers-3159162

HPV and lung cancer: a systematic review by Sequeira et al summarizes the role of human papillomavirus (HPV) in lung cancer development. This systematic review aims to explore 20 the diagnostic criteria, epidemiology, aetiology, and prognosis of HPV infection in lung cancer. A total of 97 studies encompassing 9,098 patients worldwide, revealing varied HPV infection rates in lung cancer, ranging from 0% to 69%, were analyzed. While HPV16/18 was predominant in some regions, its association with lung cancer remained inconclusive due to conflicting findings. Some studies suggested a limited role of HPV in lung carcinogenesis, particularly in non- smokers. Despite inconclusive evidence, intriguing associations between HPV and lung adenocarcinoma and squamous cell carcinoma have emerged. Further research with standardized methodologies and larger cohorts is needed for clarity.  The review presents a nice summary; however; the study needs additional information to gain interest among wide cancer and non-cancer researches.

Major comment:

1.     The authors summarize the past research.  The HPV infection deregulates p53 and pRB pathway and this known for a quite long time. The authors could summarize how to resolve this contradiction in a cartoon model and conclude their hypothesis will enhance the quality of the manuscript.

2.     The manuscript needs a strong English edit.

Reviewer 3 Report

Comments and Suggestions for Authors

1.        The plagiarism software, iThenticate, detected that this manuscript contains a 26% overlap with several published papers/literature, such as in Introduction, Methods, Results, and section 3.4 Prognosis. Specifically, the overlap came from two major papers [1,2]. The authors must rephrase and revise the draft completely to address this issue. The authors should also take immediate action to maintain the integrity and scholarly value of their work.

2.        In both Figure 1 and Figure 2, the authors claimed that HPV-DNA was integrated and detected in human lung tissues. What was the abundance or proportion of these HPV-DNAs? Was it identified using whole genome sequencing or exome sequencing?

Plagiarism sources:

(1)     From PsO to PsA: the role of TRM and Tregs in psoriatic disease, a systematic review of the literature

(2)     Human Papillomavirus Oncoproteins Confer Sensitivity to Cisplatin by Interfering with Epidermal Growth Factor Receptor Nuclear Trafficking Related to More Favorable Clinical Survival Outcomes in Non-Small Cell Lung Cancer

Comments on the Quality of English Language

Moderate editing of English language required.

Round 2

Reviewer 2 Report

Comments and Suggestions for Authors

The manuscript is improved now.

Reviewer 3 Report

Comments and Suggestions for Authors

The authors have addressed my previous comments and improved the manuscript.

Comments on the Quality of English Language

Minor editing of English language required.